# Effects of Spatial Variability and Drainage on Extracellular Enzyme Activity in Coastal Freshwater Forested Wetlands of Eastern North Carolina, USA

**Kevan J. Minick [1],\*, Maricar Aguilos [2], Xuefeng Li [2], Bhaskar Mitra [3], Prajaya Prajapati [4] and John S. King [2]**

1   Nicholas School of the Environment, Duke University, Durham, NC 27708, USA
2   Department of Forestry and Environmental Resources, North Carolina State University, Raleigh, NC 27695, USA; mmaguilo@ncsu.edu (M.A.); lxf.victor@gmail.com (X.L.); john_king@ncsu.edu (J.S.K.)
3   School of Informatics, Computing, and Cyber Systems, Northern Arizona University, Flagstaff, AZ 86011, USA; bhaskar.mitra6@gmail.com
4   Colleges of Art, Sciences and Education, Florida International University, Miami, FL 33131, USA; pprajapa@fiu.edu
\*   Correspondence: kevan.minick@duke.edu; Tel.: +1-919-630-3307

**Abstract:** Drainage of freshwater wetlands is common in coastal regions, although the effects on microbial extracellular enzyme activity (a key mediator of soil organic matter decomposition) in relation to spatial variability (microtopography and soil depth) are poorly understood. Soils were collected from organic (Oi, Oe, Oa) and mineral (A, AB, B) horizons from a natural and drained coastal forested wetland in North Carolina, USA. Activity of seven enzymes were measured: $\alpha$-glucosidase (AG), $\beta$-glucosidase (BG), cellobiohydrolase (CBH), xylosidase (XYL), phenol oxidase (POX), peroxidase (PER) and N-acetyl glucosamide (NAG). Enzyme activity rates were normalized by soil weight, soil organic C (SOC), and microbial biomass C (MBC). Specific enzyme activity (per SOC or MBC) was more sensitive to drainage and soil depth compared to normalization by soil weight. In Oi and Oa horizons, specific enzyme activity (per MBC) (AG, BG, XYL, POX, PER) was higher in the natural compared to drained wetland but lower (AG, CBH, XYL, POX, PER, NAG) in the AB or B mineral soils. Results from this study indicate that organic soil horizons of natural freshwater wetlands contain a highly active microbial community driven by inputs of plant-derived C, while deeper soils of the drained wetland exhibit higher microbial metabolic activity, which likely plays a role in SOC storage of these systems.

**Keywords:** microbial activity; wetland drainage; microtopography; decomposition; specific enzyme activity; soil carbon

## 1. Introduction

Microbial extracellular enzyme activity controls decomposition of soil organic matter [1]. Soil microorganisms produce a suite of extracellular enzymes to degrade organic carbon (C) and nitrogen (N) substrates (including $\alpha$-glucosidase, $\beta$-glucosidase, cellobiohydrolase, xylosidase, phenol oxidase, peroxidase, and N-acetyl glucosamide), to liberate energy and key nutrients (e.g., N, phosphorus, sulfur) from the soil organic matter matrix [2,3]. This has been shown to be particularly important in wetland ecosystems where saturated soils can reduce oxidative and hydrolytic enzyme activity [4,5], resulting in preservation of organic matter in soils. Of particular interest, in terms of regulation of microbial breakdown of soil organic C (SOC), is the influence of anaerobia on extracellular enzyme production. Freeman et al. [5] found that the activity of phenol oxidase, an enzyme which catalyzes the breakdown of phenolic and lignin compounds via oxygen, is suppressed under oxygen-free conditions, along with that of hydrolytic enzymes (because of inhibition due to the buildup of phenolic compounds). Importantly though, recent evidence suggests

freshwater wetlands with fluctuating water tables [6] may remain oxygenated and support microbial activity that has previously been thought to only occur under strictly anaerobic soil conditions [7–9].

Wetland drainage is common practice in coastal regions of the southeastern US, particularly for the purpose of management for pine silviculture [10]. Drainage of wetlands reduces water table depth and increases soil aeration, which typically results in enhanced SOC decomposition [11,12], although this is not always the case [13–15]. At the Alligator River National Wildlife Refuge in coastal North Carolina, the high water table in these predominantly freshwater forested wetland has preserved organic matter in a less decomposed state compared to an adjacent drained wetland site [16]. Thus, it would be expected that drainage of wetlands influences microbial enzyme activity as it is a key rate limiting step to decomposition of soil organic matter [17]. The response of enzyme activity to drainage has been shown to depend on numerous environmental factors such as the concentration of phenolic and humic compounds, soil pH, $O_2$ levels, and the type of enzyme measured [4,5,15,18–21]. For instance, enzyme activity has been shown to: decrease in the presence of phenolic and humic compounds [19,20], increase with increasing soil pH [15] and increase with increasing $O_2$ levels (particularly oxidative enzymes) [5]. Particularly, phenol oxidase activity has been shown to not change [4] or decrease [15], while hydrolytic enzymes will typically increase in response to drainage. This suggests that the microbial response to drainage is related to site specific characteristics and the type of enzyme measured [5,15].

Surface spatial heterogeneity may also play an important role in microbial processes in both natural and drained wetland sites. Microsites unique to wetlands (hummock-hollow microtopography) are influential on soil C and nutrient cycles, microbial community structure and function, and belowground plant processes [6,16,22–24]. Hummocks in non-tidal freshwater wetlands (such as those in this study) tend to have greater C and nutrient cycling rates [6,16], respiration rates [25], and are the primary location of tree growth and fine-root biomass compared to hollows [24,26,27]. Below the surface organic horizons though, organic and mineral soils from both hummocks and hollows exhibit similar saturation levels and have similar bulk SOC isotopic signatures and MBC [16]), indicating that these subsurface horizons may not differ in C quality or microbial activity. Managed pine plantations on drained wetland sites also exhibit a "man-made" form of microtopography which has also been shown to be important for both C cycling and storage [28,29] and N cycling and transformations [30]. These wetlands are specifically drained for intensively managed pine plantations characterized by localized high spots and low spots due to site preparation which involves the formation of raised beds [31]. Raised beds, where trees are planted, increase seedling survival due to enhanced soil aeration and reduced competition from herbaceous and woody vegetation as young forests aggrade. Between the rows of beds, interbed regions exist which are lower in elevation compared to beds and are typically unmanaged zones within these forested plantations. To our knowledge, microbial enzyme activity has not been measured in relation to microtopographic features within drained wetlands under intensively managed pine silviculture.

Soil depth is another type of spatial variability that influence microbial processes and soil C dynamics. In general, SOC and microbial biomass C (MBC) decreases with soil depth in upland and wetland soils [16,32–34]. As C substrate quality and quantity available for microbial degradation are altered with soil depth, changes in extracellular enzyme activity are expected [35], but it can depend on how enzyme activity is normalized and be enzyme specific. Previous studies have indicated that enzyme activity rates calculated per g dry soil may not be comparable to the specific enzyme activity (enzyme activity per g SOC or MBC) and trends in enzyme activity with soil depth will vary between enzyme type and how enzyme activity is normalized [33,35]. In one experiment, activity of peroxidase per unit mass of organic matter increased with depth while all other enzymes measured decreased (including β-glucosidase, cellobiohydrolase, xylosidase, N-acetylglucosaminidase, phenol oxidase and acid phosphatase) (Malaysian Peat Swamp Forest; [36]). Other studies

found that α-glucosidase and acid phosphatase per unit mass of MBC increased with soil depth while β-glucosidase, xylosidase, cellobiohydrolase and N-acetyl glucosaminidase had no change with depth (Luquillo Critical Zone Observatory in northeastern Puerto Rico; [33]). Finally, Webster et al. [34] found that in a hardwood swamp in the Canadian Great Lakes–St. Lawrence forest region, enzyme activity rates expressed per unit mass of dry soil (including β-glucosidase, cellobiohydrolase, N-acetylglucosaminidase, sulfatase and acid phosphatase) decreased with depth while no change with depth was detected for phenol oxidase or peroxidase activity. However, few studies have measured specific enzyme activity (normalized by both SOC and MBC) in conjunction with enzyme rates normalized by soil weight, and over such a wide range of soil organic and mineral horizons in these wetlands. Estimating specific enzyme activity may provide a more sensitive metric to detect changes in soil microbial activity and soil organic matter transformations under different land-use and with soil depth, as it accounts for changes in SOC or MBC content [35]. Not surprisingly, spatial heterogeneity in soil chemistry (e.g., pH, electron acceptor concentration), $O_2$ levels, soil microbial communities, and C substrates (e.g., presence of phenolics or humics, lignin concentrations) with soil depths also affects the types and activity of enzymes within the soil [19,21,37–39] and can be site specific.

In the current study, our objective was to quantify microbial extracellular enzyme activity rates targeting a suite of SOC substrates, as well as that of one N-degrading enzyme, in natural and drained freshwater forested wetlands in a temperate region of coastal North Carolina, USA. Specifically, we investigated the effects of wetland drainage and spatial variability (microtopography and soil depth) on enzyme activity and if patterns in enzyme activity differed depending on how enzyme activity was normalized. We hypothesized that: (1) enzyme activity would be higher in the drained wetland compared to the natural wetland; (2) enzyme activity would decrease with soil depth when normalized per unit mass of dry soil and show a variable response when normalized per unit mass of SOC or MBC; and (3) microtopography would be a major determinant of enzyme activity in the natural wetland but not drained wetland.

## 2. Methods

### 2.1. Site Description

The natural forested wetland was located at Alligator River National Wildlife Refuge (ARNWR) in Dare County, North Carolina (35°47′ N, 75°54′ W), with the Alligator River to the west, Albemarle Sound to the north, and the Croatan Sound to the east (Supplementary Figure S1). The refuge was established in 1984 and is characterized by a diverse assemblage of pocosin (wetland fed by rainwater or groundwater, acidic peat and mineral soils, oligotrophic, woody vegetation) wetland types and swamp forests [40]. ARNWR has a network of roads and canals, but in general contains vast expanses of minimally disturbed forested- and shrub-wetlands. Thirteen plots were established in a 4 km$^2$ area in the middle of a bottomland hardwood forest surrounding a 35-m eddy covariance flux tower (US-NC4 in the AmeriFlux database) [41]. Of the 13 plots (7 m radius), the five central plots were utilized for this study. Over-story plant species composition was predominantly composed of black gum (*Nyssa sylvatica*), swamp tupelo (*Nyssa biflora*), bald cypress (*Taxodium distichum*), with occasional red maple (*Acer rubrum*), sweet gum (*Liquidambar styraciflua*), white cedar (*Chamaecyparis thyoides*), and loblolly pine (*Pinus taeda*). The understory was predominantly fetterbush (*Lyonia lucida*), bitter gallberry (*Ilex glabra*), red bay (*Persea borbonia*), and sweet bay (*Magnolia virginiana*). The mean annual temperature and precipitation from an adjacent meteorological station (Manteo AP, NC, 35°55′ N, 75°42′ W, National Climatic Data Center) for the period 1981–2010 were 16.9 °C and 1270 mm, respectively. These wetlands are characterized by a hydroperiod that operates over short time scales and is driven primarily by variation in precipitation. Soils are classified as Pungo series histosols (very poorly drained dystic thermic typic Haplosaprist) with a deep, highly decomposed muck layer overlain by a shallow, less decomposed peat layer and underlain by highly reduced mineral sediments of Pleistocene origin [42]. Ground elevation is <1 m above sea level.

The drained wetland was an intensively managed loblolly pine plantation located in Washington County, NC (35°48′ N, 76°40′ W) within the lower coastal plain mixed forest province (Supplementary Figure S1). In the late 1800's/early 1900's, the area was clearcut and ditched for agriculture. Since the mid-1960's, this site has been owned and operated by Weyerhaeuser NR Company for commercial timber production and is currently in its fourth rotation of loblolly pine. The pine plantation watersheds are managed with a network of parallel ditches (0.9–1.3 m deep; 90 m spacing) and more widely spaced roadside canals. Drainage lowers the height of the water table and improves site access and tree productivity by reducing stresses caused by excessive soil water, especially during winter months. The current loblolly pine stand is also part of the AmeriFlux core sites (US-NC2 in the AmeriFlux database) and was planted in 1992, on bedded rows, with 2-year-old half sibling seedlings after clear-cutting the previous mature pine plantation. Planting spacing was 1.5 m by 4.5 m. Bedding is common best management practice along the southeastern lower coastal plain [31] in order to increase growth and survival of seedlings. Bedding creates unique micro-topography in managed pine forests, resulting in high (bed) and low (interbed; space between beds) micro-topographic locations, which influence SOC stabilization [28] and N cycling [43]. Thirteen plots were established in a mature loblolly pine stand surrounding a 28-m eddy flux tower [41]. Of these 13 plots (7 m radius), five central plots were utilized for this study. The understory was primarily composed of young red maple (*A. rubrum*) and other non-woody plants such as devil's walking stick (*Aralia spinosa*), pokeweed (*Phytolacca americana*), beautyberry (*Callicarpa americana*), giant cane (*Arundinaria macrosperma*) and meadow grass (*Poa* spp.). The histic-mineral soil at this site is classified as a Belhaven series histosol (loamy, mixed, dystic, thermic Terric Haplosaprists) with a highly decomposed organic matter layer underlain by loamy marine sediments. Belhaven and Pungo soil series are typically found within the same soil map of this region, with Belhaven series soils having somewhat higher permeability due to the loamy soil texture [44]. The soils are very poorly drained with a ground elevation <5 m above sea level. The long-term (1945–2010) average annual precipitation was 1290 ± 199 mm, evenly distributed throughout the year. Long term mean annual temperature averaged 15.5 °C, with a monthly high temperature occurring in July (26.6 °C), and a monthly low occurring in January (6.4 °C). Within each site we have a very large spatial distribution of plots (>500 × 500 m$^2$) which captures the full range of spatial variation (in vegetation, soils, hydrology) that exists within these types of ecosystems.

### 2.2. Soil Sample Collection and Processing

In the natural wetland, two soil samples were collected from high (hummock) and low (hollow) micro-topographic positions in early 2016 within each of the five plots (Supplementary Figure S2A). First, surface litter (Oi horizon, horizon depth = 2 ± 0.02 cm) was collected within a 0.25 × 0.25 m$^2$ quadrant (Supplementary Figure S3A). In the natural wetland, rooting density is very high at the surface [16,24] rendering traditional coring methods ineffective. After surface litter was collected, a saw was used to cut a 0.25 × 0.25 m$^2$ monolith (Oe horizon, horizon depth = 8 ± 0.04 cm). Below the Oe horizon was an approximate 20 cm deep space containing very large coarse roots and water and/or air. This was particularly evident in hummocks. Finally, the organic (Oa horizon, horizon depth = 44 ± 3.5 cm) and A (horizon depth = 10 ± 1.7 cm), AB (horizon depth = 11 ± 2.1 cm) and B (top 10 cm) mineral soil horizons below the root mat were sampled with a Macaulay auger (Supplementary Figure S3A).

In the drained wetland, two soil samples were collected from high (bed) and low (interbed) micro-topographic positions in early 2016 from each of the five plots (Supplementary Figure S2B). A dense surface rooting zone was not present at the drained wetland as at the natural wetland. Therefore, a 5 cm diameter by 1.5 m long PVC corer was used to sample organic and mineral soils. Surface litter samples (Oi horizon, horizon depth = 2 ± 0.02 cm) were first collected from a 0.25 × 0.25 m$^2$ quadrant (Supplementary Figure S3B). Organic (Oe and Oa) and mineral (A, AB, and B) soil horizons were then

sampled beneath the collected litter using the PVC corer and subdivided into the same horizons as the natural wetland (Supplementary Figure S3B): Oe organic horizon (horizon depth = $4 \pm 0.5$ cm), Oa organic horizon (horizon depth = $19 \pm 2.1$ cm), A mineral horizon (horizon depth = $10 \pm 1.2$ cm), AB mineral horizon (horizon depth = $7 \pm 1.3$ cm), and top 10 cm of B mineral horizons.

### 2.3. Soil pH, Volumetric Water Content, Eh, and Percent Total Roots

Live and dead roots were removed from the samples and soil was removed from roots by rinsing in distilled water (dH$_2$O). Roots were then dried to constant mass and ground to a fine powder in a Wiley mill. Ground root samples were analyzed in duplicate for C concentration on a Picarro G2201-*i* Cavity Ring-Down Spectroscopy (CRDS) isotopic CO$_2$/CH$_4$ analyzer (Picarro Inc., Sunnyvale, CA, USA) interfaced with a Picarro Combustion Module (Picarro Inc., Sunnyvale, CA, USA). After roots were removed, the remaining fresh soil from each of the two samples per plot and horizon were composited and gently homogenized by hand. Soil pH was measured on fresh soil samples with a glass electrode in a 1:2mixture (by mass) of soil and dH$_2$O. Gravimetric soil water content was determined by drying a fresh subsample for 24 h at 105 °C. Soil redox potential (Eh = mV) was measured on fresh soil samples in the laboratory using a Martini ORP 57 ORP/_C/_F meter (Milwaukee Instruments, Inc., Rocky Mount, NC, USA).

### 2.4. Soil Organic Carbon and Soil Microbial Biomass

Data for SOC and MBC were previously published in an experiment utilizing the same sampling date and processing techniques used for this study (see [16]). Briefly, SOC concentration was determined on dried (70 °C), homogenized and finely ground soils following MBC and enzyme analysis. SOC concentration was quantified in duplicate on a Picarro G2201-*i* Cavity Ring-Down Spectroscopy (CRDS) isotopic CO$_2$/CH$_4$ analyzer (Picarro Inc., Sunnyvale, CA, USA) interfaced with a Picarro Combustion Module (Picarro Inc., Sunnyvale, CA, USA). The chloroform fumigation extraction (CFE) method was adapted from Vance et al. [45] in order to estimate MBC. For more information regarding the methods of this experiment see Minick et al. [16].

### 2.5. Extracellular Enzyme Activity and Specific Enzyme Activity Calculations

A subset of each fresh, root-free, homogenized soil from each site, sampling location and horizon was frozen at −20 °C until enzyme analysis. Soils were analyzed for a suite of C-degrading enzymes: α-glucosidase (AG; EC 3.2.1.20), β-glucosidase (BG; EC 3.2.1.21), xylosidase (XYL; EC 3.2.1.37), cellobiohydrolase (CBH; EC 3.2.1.91), phenol oxidase (POX; EC 1.10.3.2), and peroxidase (PER; EC 1.11.1.7). Soils were also analyzed for the N-degrading enzyme N-acetyl glucosamide (NAG; EC: 3.2.1.30). Substrates for all enzyme assays were dissolved in 50 mM, pH 5.0 acetate buffer solution.

Hydrolytic enzymes (AG, BG, XYL, CBH, and NAG) were measured using techniques outlined in Sinsabaugh et al. [46]. Approximately 0.5 g dry weight of soil sample was suspended in 50 mL of a 50 mM, pH 5.0 acetate buffer solution and homogenized in a blender for 1 min. In a 2 mL centrifuge tube, 0.9 mL aliquot of the soil-buffer suspension was combined with 0.9 mL of the appropriate 5 mM p-nitrophenyl substrate solution for a total of three analytical replicates. Additionally, duplicate background controls consisted of 0.9 mL aliquot of soil-buffer suspension plus 0.9 mL of acetate buffer and four substrate controls were analyzed consisting of 0.9 mL substrate solution plus 0.9 mL buffer. The samples were agitated for 2–6 h at the treatment temperature. Samples were then centrifuged at 8160 g for 3 min. Supernatant (1.5 mL) was transferred to a 15 mL centrifuge tube containing 150 μL 1.0 M NaOH and 8.35 mL dH$_2$O. The resulting mixture was vortexed and a subsample transferred to a cuvette and the optical density at 410 nm was measured on a spectrophotometer (Beckman Coulter DU 800 Spectrophotometer, Brea, CA, USA).

Oxidative enzymes (POX and PER) were measured using techniques outlined in Sinsabaugh et al. [47]. POX and PER are primarily involved in oxidation of phenol compounds and depolymerization of lignin. The same general procedure for hydrolytic enzymes was followed utilizing a 5 mM L-3,4-Dihydroxyphenylalanine (L-DOPA) (Sigma-Aldrich Co. LLC, St. Louis, MO, USA) solution as the substrate. After set up of analytical replicates and substrate and background controls, the samples were agitated for 2–4 h at the treatment temperature. Samples were then centrifuged at 8160 g for 3 min. The resulting supernatant turns an intense indigo color. PER was determined similar to that of POX, except with the addition of 0.2 mL of 0.3% $H_2O_2$ to all sample replicates and controls. Supernatant (1.4 mL) was transferred directly to a cuvette and the optical density at 460 nm was measured on the spectrophotometer.

For all enzymes, the mean absorbance of two background controls and four substrate controls was subtracted from that of three analytical replicates and divided by the molar efficiency (1.66/μmol), length of incubation (h), and soil dry weight. PER activity was determined by subtracting the absorbance for POX from the absorbance for PER. Enzyme activity at each measurement date was expressed as μmol substrate per unit SOC per hour (μmol g SOC $^{-1}$ h$^{-1}$), per unit dry weight of soil and time (μmol g$^{-1}$ h$^{-1}$), and per unit MBC and time (μmol g MBC $^{-1}$ h$^{-1}$). The SOC pool size and MBC were measured in a previous study using the same soils and details on this analysis can be found in Minick et al. [16].

### 2.6. Statistical Analysis

All data were analyzed using three-way ANOVA (PROC GLM). Site, microsite, and soil horizon were treated as fixed effects. Data were log transformed where necessary to meet assumptions of ANOVA. All data were plotted as means of raw data. If significant main effects or interactions were identified in the three-way ANOVA ($p < 0.05$), then differences were tested using least square means post-hoc analysis. All ANOVA based statistical analyses were performed using SAS 9.4 software (SAS Institute, Cary, NC, USA).

The Principal Component Analysis was performed using the *res.pca* function from *FactoMineR* package in R. Correlations between environmental variables and enzyme activities were analyzed using the *PerformanceAnalytics* and *corrplot* packages. Both analyses were processed in R version 4.1.1.

### 3. Results

### 3.1. The Enzyme Activity in Different Soil Horizons and Microsites

The natural wetland was highly acidic, with pH ranging from 4.2–4.9 and minimal differences between soil horizons (Table 1). The drained wetland was also acidic, with pH ranging from 4.0–5.6 and differences between soil horizons primarily driven by higher pH in Oi and B horizons (Table 1). The Oi and B horizon pH were higher in the drained wetland compared to the natural wetland. In the natural wetland, soil water content was higher in all soil horizons compared to the drained wetland (Table 1). Not surprisingly, hollows in the natural wetland had higher water content in the Oi and Oe horizon compared to that of the hummocks (Table 1).

Site and soil horizon effects were the most important factors influencing enzyme activity regardless of how the data were normalized, while microsite had little to no effect (Tables 2–4). Decreases in enzyme activity with soil depth were most pronounced for those normalized by soil weight (Figure 1), with the highest rates found at the surface in Oi and Oe soil. Data normalized by SOC (Figure 2) and MBC (Figure 3) exhibited more similar depth patterns than those normalized by soil weight (Figure 1). For enzyme rates normalized by SOC (Figure 2) and MBC (Figure 3), the highest rates were typically found in the Oi horizon or in the B mineral horizon where XYL, POX, and PER enzyme rates were similar to or higher than those measured in the Oi (Figures 2 and 3).

**Table 1.** Soil pH and water content (%) for samples collected from organic and mineral horizons of a natural coastal freshwater forested wetland (Natural Wetland) and an intensively managed loblolly pine plantation (Drained Wetland) in coastal North Carolina, USA. Data represent means with standard error ($n$ = 5). Significant differences between soil horizons at each site and sampling location are indicated by superscript upper-case letters ($p < 0.05$). Significant differences between each site and micro-topographic location (e.g., hummock, hollow, bed, or interbed) at each soil horizon are indicated by superscript lower-case letters ($p < 0.05$).

| | pH | | | | Soil Water Content (%) | | | |
|---|---|---|---|---|---|---|---|---|
| Horizon | Natural Wetland | | Drained Wetland | | Natural Wetland | | Drained Wetland | |
| | Hummock | Hollow | Bed | Interbed | Hummock | Hollow | Bed | Interbed |
| Oi | 4.8 ± 0.1 [Ab] | 4.3 ± 0.1 [Bb] | 5.6 ± 0.3 [Aa] | 5.6 ± 0.2 [Aa] | 67 ± 3 [Bb] | 90 ± 1 [Aa] | 48 ± 5 [BCc] | 55 ± 5 [BCc] |
| Oe | 4.2 ± 0.1 [Ba] | 4.3 ± 0.1 [Ba] | 4.2 ± 0.2 [DCEa] | 4.5 ± 0.2 [CDEa] | 85 ± 1 [Ab] | 90 ± 1 [Aa] | 66 ± 2 [Ad] | 78 ± 2 [Ac] |
| Oa1 | 4.4 ± 0.1 [ABa] | 4.4 ± 0.1 [Ba] | 4.0 ± 0.1 [Eb] | 4.3 ± 0.1 [Ea] | 84 ± 4 [Aa] | 86 ± 2 [Aa] | 61 ± 1 [Ab] | 64 ± 3 [Bb] |
| Oa2 | 4.3 ± 0.1 [ABa] | 4.4 ± 0.2 [ABa] | 4.2 ± 0.1 [DEa] | 4.4 ± 0.1 [DEa] | 78 ± 6 [Aa] | 74 ± 5 [Ba] | 60 ± 3 [Ab] | 61 ± 1 [BCb] |
| Oa3 | 4.4 ± 0.1 [ABb] | 4.4 ± 0.2 [Bb] | 4.6 ± 0.2 [Cab] | 4.7 ± 0.0 [BCa] | 74 ± 5 [ABa] | 71 ± 6 [BCa] | 45 ± 5 [Cb] | 50 ± 5 [Cb] |
| A | 4.4 ± 0.2 [ABa] | 4.4 ± 0.2 [ABa] | 4.5 ± 0.2 [DCa] | 4.7 ± 0.1 [BCDa] | 60 ± 4 [BCa] | 59 ± 5 [Ca] | 31 ± 3 [Db] | 29 ± 3 [Db] |
| AB | 4.5 ± 0.3 [ABa] | 4.5 ± 0.3 [ABa] | 5.0 ± 0.1 [Ba] | 5.0 ± 0.1 [ABa] | 52 ± 6 [Ca] | 45 ± 4 [Da] | 20 ± 2 [DEb] | 24 ± 3 [DEb] |
| B | 4.7 ± 0.2 [ABb] | 4.9 ± 0.3 [Aab] | 5.4 ± 0.2 [ABa] | 5.4 ± 0.2 [Aa] | 36 ± 1 [Da] | 32 ± 1 [Ea] | 21 ± 1 [Eb] | 20 ± 1 [Eb] |

**Table 2.** Results (F-values and significance) of three-way ANOVA testing effects of site (natural vs. drained wetland), microsite (high vs. low from each site), and soil horizon on extracellular enzyme activity per unit mass of soil ($\mu mol\ g\ soil^{-1}\ h^{-1}$) measured in soils from organic and mineral soil horizons. Bold and italicized values represent the highest level significant main effect or interaction, upon which further interpretation was based.

| Source | AG | BG | CBH | XYL | POX | PER | NAG |
|---|---|---|---|---|---|---|---|
| Site | 16.1 *** | 1.2 | 2.42 | 0.7 | 1.5 | 45.5 *** | 14.6 *** |
| Microsite | 1.8 | 0.3 | 0.0 | 0.4 | 1.0 | 0.0 | 1.8 |
| Horizon | 194.3 *** | ***394.2*** | 276.6 *** | 235.3 *** | ***102.2*** *** | 49.6 *** | 198.2 *** |
| Site * Microsite | 0.1 | 1.2 | 5.6 * | 1.1 | 0.8 | 0.0 | 3.6 |
| Site * Horizon | *15.3* *** | 0.3 | 1.4 | *2.8* * | 1.0 | 10.8 *** | *12.4* *** |
| Microsite * Horizon | 1.4 | 0.8 | 0.2 | 0.4 | 0.7 | 2.5 * | 0.5 |
| Site * Microsite * Horizon | 0.3 | 0.4 | *2.8* * | 1.5 | 0.5 | *3.5* ** | 1.4 |

\* $p < 0.05$, \*\* $p < 0.01$, \*\*\* $p < 0.001$. Activity of seven enzymes were measured: α-glucosidase (AG), β-glucosidase (BG), cellobiohydrolase (CBH), xylosidase (XYL), phenol oxidase (POX), peroxidase (PER) and N-acetyl glucosamide (NAG).

**Table 3.** Results (F-values and significance) of three-way ANOVA testing effects of site (natural vs. drained wetland), microsite (high vs. low from each site), and soil horizon on extracellular enzyme activity per unit mass of SOC ($\mu mol\ g\ SOC^{-1}\ h^{-1}$) measured in soils from organic and mineral soil horizons. Bold and italicized values represent the highest level significant main effect or interaction, upon which further interpretation was based.

| Source | AG | BG | CBH | XYL | POX | PER | NAG |
|---|---|---|---|---|---|---|---|
| Site | 0.8 | 0.1 | 10.2 ** | 5.8 ** | 0.4 | 0.6 | 23.3 *** |
| Microsite | 0.8 | 0.1 | 0.1 | 0.4 | 0.8 | 0.0 | 1.0 |
| Horizon | 106.7 *** | ***216.1*** *** | 249.8 *** | 41.1 *** | ***28.8*** *** | 16.1 *** | 107.7 *** |
| Site * Microsite | 0.2 | 0.5 | *4.6* * | 0.1 | 0.2 | 0.1 | *6.5* * |
| Site * Horizon | *13.3* *** | 0.3 | *2.3* * | *2.3* * | 1.1 | *12.4* *** | *5.4* ** |
| Microsite * Horizon | 0.8 | 0.7 | 0.2 | 0.3 | 0.3 | 1.1 | 0.8 |
| Site * Microsite * Horizon | 0.3 | 0.5 | 2.1 | 0.8 | 0.5 | 1.8 | 0.4 |

\* $p < 0.05$, \*\* $p < 0.01$, \*\*\* $p < 0.001$. Activity of seven enzymes were measured: α-glucosidase (AG), β-glucosidase (BG), cellobiohydrolase (CBH), xylosidase (XYL), phenol oxidase (POX), peroxidase (PER) and N-acetyl glucosamide (NAG).

**Table 4.** Results (F-values and significance) of three-way ANOVA testing effects of site (natural vs. drained wetland), microsite (high vs. low from each site), and soil horizon on extracellular enzyme activity per unit mass of MBC ($\mu$mol g MBC$^{-1}$ h$^{-1}$) measured in soils from organic and mineral soil horizons. Bold and italicized values represent the highest level significant main effect or interaction, upon which further interpretation was based.

| Source | AG | BG | CBH | XYL | POX | PER | NAG |
|---|---|---|---|---|---|---|---|
| Site | 2.8 | 0.0 | 2.8 | 0.7 | 0.5 | 6.2 ** | 15.2 *** |
| Microsite | 1.0 | 0.0 | 0.9 | 0.2 | 0.0 | 0.5 | 0.1 |
| Horizon | 11.2 *** | 54.8 *** | 71.4 *** | 3.0 * | 4.3 ** | 32.0 *** | 23.3 *** |
| Site * Microsite | 1.2 | 0.1 | 0.5 | 0.9 | 0.0 | 0.8 | 1.7 |
| Site * Horizon | *10.3 *** | *5.0 ** | *4.2 ** | *4.4 ** | *3.5 ** | *11.5 *** | *4.5 ** |
| Microsite * Horizon | 1.7 | 1.2 | 0.9 | 1.6 | 0.9 | *2.9 * | 0.9 |
| Site * Microsite * Horizon | 0.7 | 0.4 | 1.3 | 0.2 | 0.4 | 1.5 | 0.8 |

* $p < 0.05$, ** $p < 0.01$, *** $p < 0.001$. Activity of seven enzymes were measured: $\alpha$-glucosidase (AG), $\beta$-glucosidase (BG), cellobiohydrolase (CBH), xylosidase (XYL), phenol oxidase (POX), peroxidase (PER) and N-acetyl glucosamide (NAG).

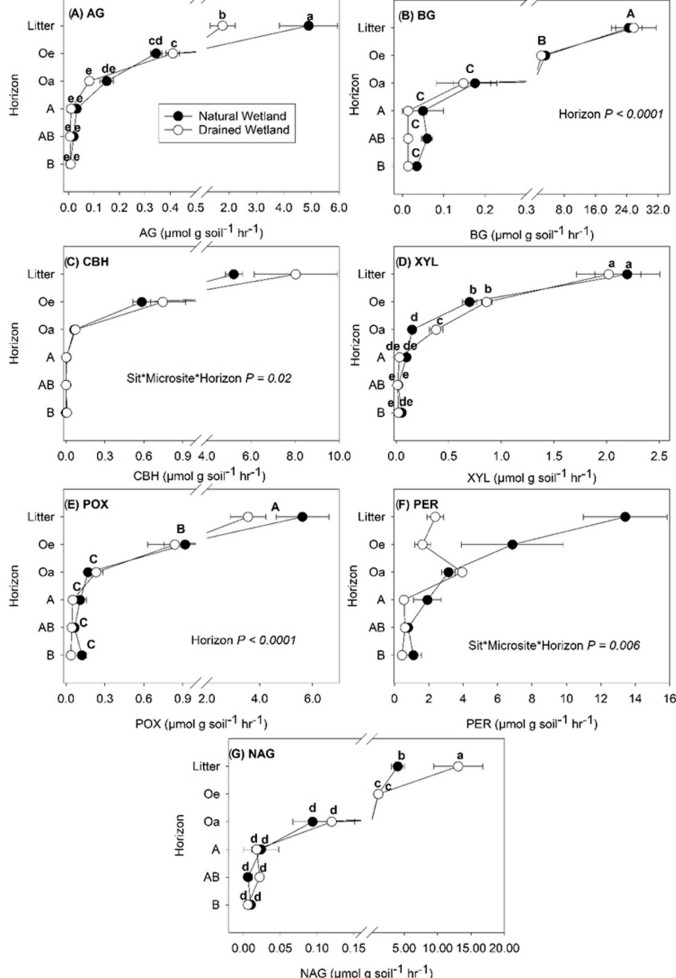

**Figure 1.** Enzyme activity rates calculated per unit mass of dry soil ($\mu$mol g soil$^{-1}$ h$^{-1}$) in soils collected from different soil horizons within a natural and managed freshwater forested wetland. Activity of seven enzymes were measured: (**A**) $\alpha$-glucosidase (AG), (**B**) $\beta$-glucosidase (BG), (**C**) cellobiohydrolase (CBH), (**D**) xylosidase (XYL), (**E**) phenol oxidase (POX), (**F**) peroxidase (PER) and (**G**) N-acetyl glucosamide (NAG). Values are means with standard error (*n* = 5) and averaged across microsite, depicting the site by horizon interaction found for most enzymes assayed. Lowercase letters represent significant differences (*p* < 0.05) between each site and horizon when a significant site by soil horizon interaction was found. Uppercase letters represent significant differences between soil horizons when a soil horizon main effect only was found.

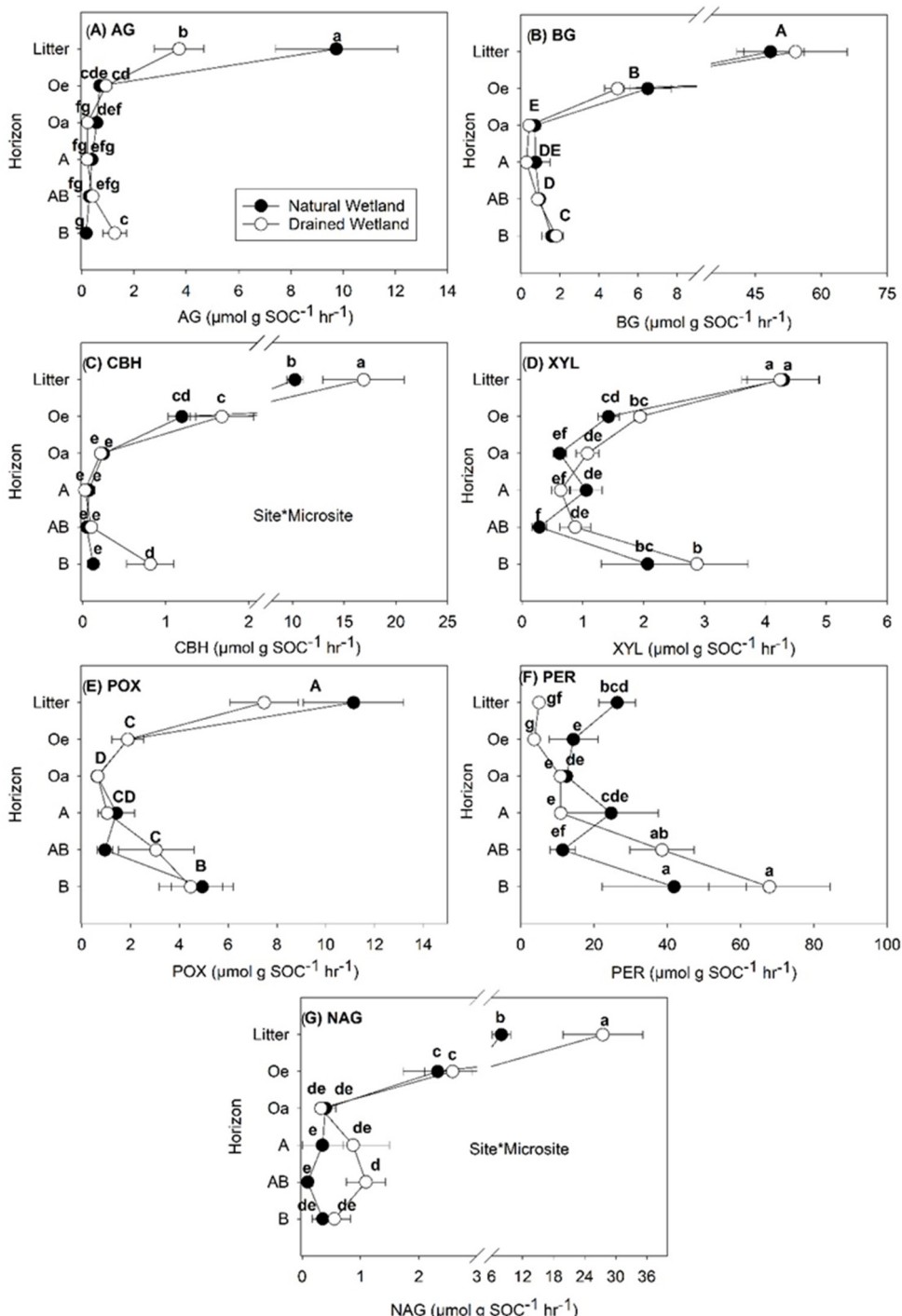

**Figure 2.** Specific enzyme activity rates calculated per unit mass of soil organic carbon (SOC) ($\mu$mol g SOC$^{-1}$ h$^{-1}$) in soils collected from different soil horizons within a natural and managed freshwater forested wetland. Activity of seven enzymes were measured: (**A**) $\alpha$-glucosidase (AG), (**B**) $\beta$-glucosidase (BG), (**C**) cellobiohydrolase (CBH), (**D**) xylosidase (XYL), (**E**) phenol oxidase (POX), (**F**) peroxidase (PER) and (**G**) N-acetyl glucosamide (NAG). Values are means with standard error ($n = 5$) and averaged across microsite, depicting the site by horizon interaction found for most enzymes assayed. Lowercase letters represent significant differences ($p < 0.05$) between each site and horizon when a significant site by soil horizon interaction was found. Uppercase letters represent significant differences between soil horizons when a soil horizon main effect only was found.

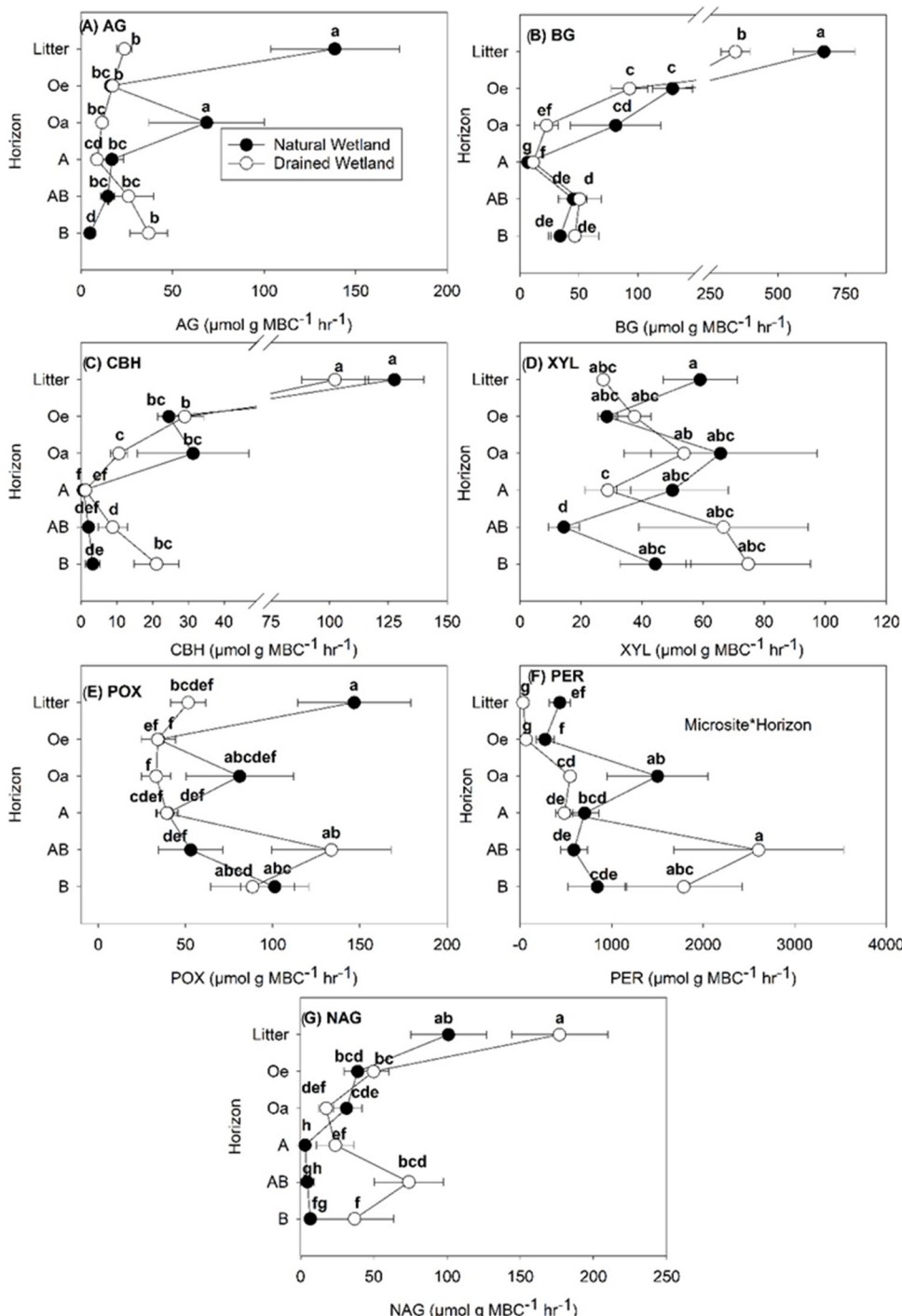

**Figure 3.** Specific enzyme activity rates calculated per unit mass of microbial biomass carbon (MBC) ($\mu$mol g MBC$^{-1}$ h$^{-1}$) in soils collected from different soil horizons within a natural and managed freshwater forested wetland. Activity of seven enzymes were measured: (**A**) $\alpha$-glucosidase (AG), (**B**) $\beta$-glucosidase (BG), (**C**) cellobiohydrolase (CBH), (**D**) xylosidase (XYL), (**E**) phenol oxidase (POX), (**F**) peroxidase (PER) and (**G**) N-acetyl glucosamide (NAG). Values are means with standard error ($n = 5$) and averaged across microsite, depicting the site by horizon interaction found for most enzymes assayed. Lowercase letters represent significant differences ($p < 0.05$) between each site and horizon when a significant site by soil horizon interaction was found. Uppercase letters represent significant differences between soil horizons when a soil horizon main effect only was found.

The enzymes AG and BG target breakdown of C compounds with short turnover rates (e.g., glucose). For AG, activity rates were higher in the Oi horizon of the natural compared to the drained wetland regardless of how the data were normalized (Figures 1A, 2A and 3A). For AG normalized by MBC, AG activity was also higher in the Oa layer of the natural wetland compared to the drained wetland (Figure 3A). In the B mineral horizon of the drained wetland, higher AG activity normalized by SOC and MBC was found compared to the natural wetland (Figures 2A and 3A), but much more pronounced for AG activity normalized by MBC with 649% higher activity compared to only 105 % higher activity for AG normalized by SOC. Activity of BG normalized by MBC was higher in the Oi and Oa soil horizons of the natural wetland compared to the drained wetland (Figure 3B). Overall, a small but significant increase in BG activity normalized by SOC and MBC was observed in the AB and/or B mineral soil horizons of both sites (Figures 2B and 3B).

The enzymes CBH and XYL target breakdown of C compounds with intermediate turnover rates (e.g., cellulose and hemicellulose). Cellobiohydrolase activity generally decreased with depth (Figures 1C, 2C and 3C), except in the B horizon where CBH activity normalized by SOC and MBC was higher in the drained wetland and similar to that of surface organic horizons excluding the Oi (Figures 2C and 3C). A three-way interaction was found between site, microsite, and horizon for CBH rates normalized by soil weight (Table 2), with CBH activity highest in the low microsite of the drained wetland and lowest in the low microsite of the natural wetland (data not shown). Activity of CBH normalized by SOC was higher in the Oi of the drained wetland (Figure 2C), while CBH activity normalized by MBC was higher in the Oa horizon of the natural wetland (Figure 3C). The activity of XYL across soil depths showed the greatest variability in trends depending on how the data were normalized (Figures 1D, 2D and 3D), particularly for XYL normalized by MBC (Figure 3D). For XYL normalized by soil weight, trends were similar to other enzymes, with rates dropping quickly to very low levels (Figure 1D). Similarly, XYL activity normalized by SOC decreased with soil depth except for that in the B horizon where XYL rates were equal to or greater than those in other soil horizons, except for Oi (Figure 2D). Activity of XYL normalized by SOC was higher in the AB mineral soil horizon of the drained compared to natural wetland (Figure 2D). Activity of XYL normalized by MBC was highly variable with soil depth and exhibited higher activity in the Oi horizon of the natural wetland and lower activity in the AB mineral soil horizon compared to the drained wetland (Figure 3D).

Both POX and PER target the breakdown of C compounds with slow turnover time (e.g., phenols and lignin, respectively). Phenol oxidase activity decreased with soil depth for data normalized by soil weight (Table 2; Figure 1E). For POX activity normalized by SOC, no effect of site was found (Table 3; Figure 2E), but did show an increase in activity in the AB and B mineral soil horizons. For POX activity normalized by MBC, higher rates were found in the Oi and lower rates in the AB horizon of the natural wetland compared to the drained wetland (Figure 3E). For PER activity normalized by soil weight, higher activity was found in the Oi, Oe, and A soil horizons of the natural wetland compared to the drained wetland and tended to decrease with depth at both sites (Figure 1F). In contrast, PER activity normalized by SOC and MBC tended to increase with soil depth (Figures 2F and 3F). Activity of PER normalized by SOC was higher in the Oi and Oe horizons and lower in the AB mineral soil horizon of the natural wetland compared to the drained wetland (Figure 2F). The activity of PER normalized by MBC was higher in the Oi, Oe, and Oa soil horizons and lower in the AB soil horizon of the natural compared to drained wetland (Figure 3F).

Finally, NAG activity tended to decrease with soil depth regardless of how enzyme rates were normalized (Figures 1G, 2G and 3G), except for the AB mineral soil horizon for data normalized by SOC and MBC in the drained wetland. In contrast to C degrading enzymes, NAG activity was higher in the Oi of the natural compared to drained wetland regardless of how data were normalized (Figures 1G, 2G and 3G). Higher activity of

NAG normalized by SOC and MBC was also measured in the AB horizon of the drained compared to natural wetland (Figure 2G). Across all soil depths, NAG rates normalized by SOC were highest in the low microsite of the drained wetland and lowest in the low microsite of the natural wetland (Table 2) (data not shown).

### 3.2. Environmental Effects on Enzyme Activity

Princiapl Component Analysis (PCA) of environmental variables (Eh, pH, total root biomass and soil water content) and enzyme activity showed different drivers of enzyme rates between the two sites and how enzyme rates were normalized, although there were some similarities (Figures 4 and 5). For enzyme activity normalized by soil weight, soil water (r = 0.37–0.86) and total root biomass (r = 0.50–0.73) showed a significant, positive correlation with all enzymes (except soil water and PER) in the natural wetland (Figures 4a and 5a). In the drained wetland, all enzymes normalized by soil weight were highly correlated with soil water content (r = 0.54–0.92), but only PER and total root biomass were significantly correlated (r = 0.30) (Figures 4b and 5b). Enzyme activity normalized per unit mass of soil was uncorrelated with pH across all enzymes in the natural wetland, while AG (r = −0.38), XYL (r = −0.48), POX (r = −0.31) and PER (r = −0.49) showed significant negative correlation in the drained wetland. Finally, POX, PER and NAG all exhibited negative correlations with Eh in the natural wetland, and all but PER in the drained wetland. For AG, BG, CBH and NAG activity normalized by SOC, soil water (r = 0.32–0.62) and total roots (r = 0.33–0.70) had a positive correlation in the natural wetland (Figures 4c and 5c). No significant correlations were found for total root biomass and enzyme activity normalized by SOC in the drained wetland, but BG, CBH and NAG did show a positive correlation with soil water content (Figures 4d and 5d). XYL, POX, and PER activity normalized by SOC all exhibited positive correlations with pH at both sites, as well as AG in the drained wetland. Similarly, enzyme activity normalized by SOC was negatively correlated with Eh in the natural (all except AG and XYL) and drained (only BG, CBH, and NAG) wetland (Figures 4 and 5). For enzyme activity normalized by MBC, significant correlations with environmental variables were minimal compared to other methods of normalization (Figure 4e,f and Figure 5e,f). AG, BG, CBH, and NAG exhibited a positive correlation with soil water content in the natural wetland (r = 0.35–0.67). In the drained wetland, BG (r = 0.41) and CBH (r = 0.45) were positively correlated with soil water content while POX (r = −0.27) and PER (r = −0.39) were negatively correlated with soil water content. For enzymes normalized by MBC, only BG (r = 0.32) and NAG (r = 0.44) were positively correlated with total root biomass in the natural wetland, and XYL (r = 0.20), POX (r = 0.28), and PER (r = 0.35) in the drained wetland. Enzyme activity normalized by MBC was uncorrelated with pH and Eh in the natural wetland. In the drained wetland, AG (r = 0.45), XYL (r = 0.31), POX (r = 0.42) and PER (r = 0.53) were positively correlated with pH and CBH (r = −0.31) was negatively correlated with Eh.

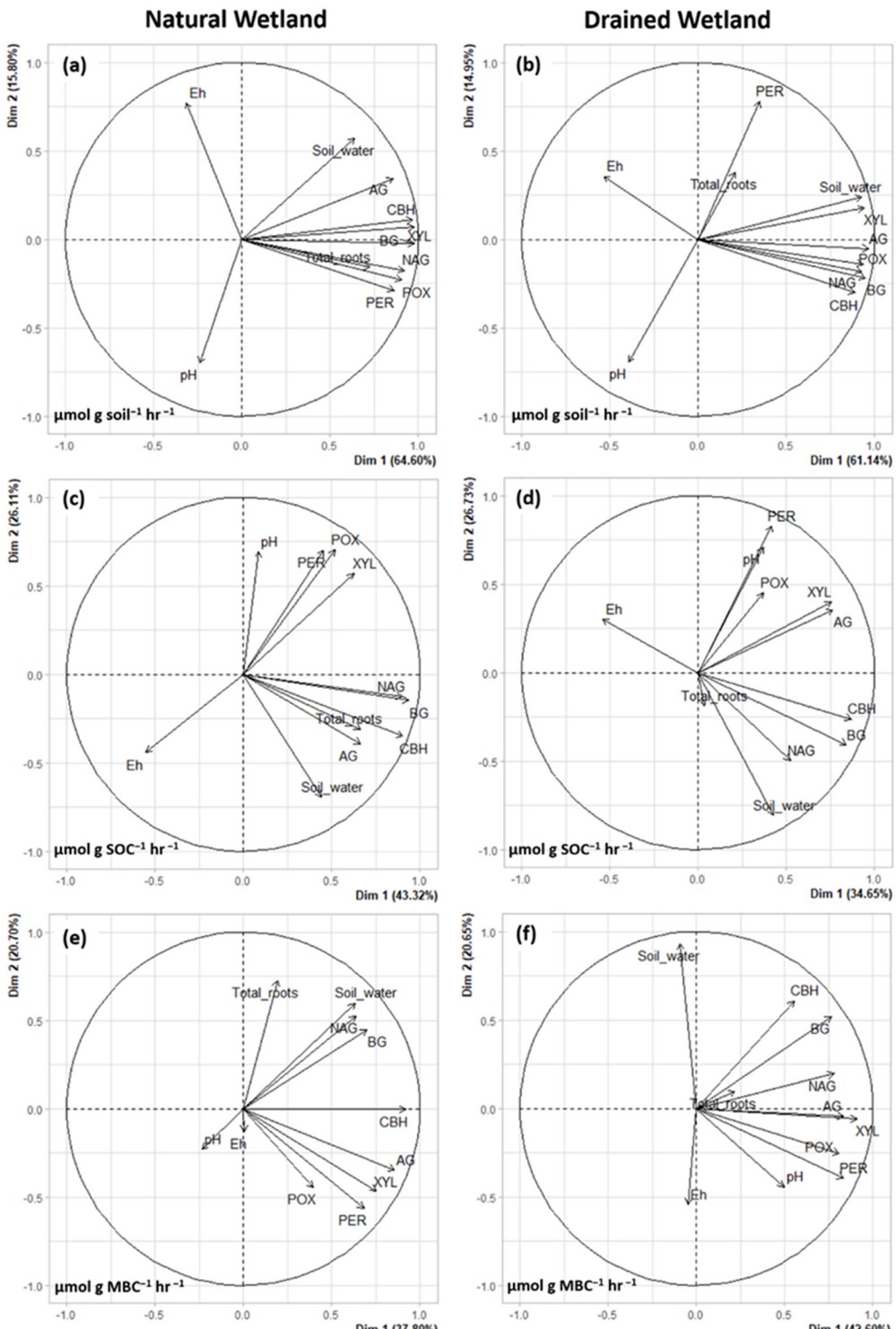

**Figure 4.** Principal Component Analysis (PCA) showing the relationship between environmental variables (pH, total root biomass, Eh, and soil water content) and extracellular enzyme activity in natural (**a,c,e**) and drained (**b,d,f**) forested wetlands in eastern North Carolina estimated using three different normalization methods (**a,b**): normalized by soil weight; (**c,d**): normalized by soil organic C content; and (**e,f**): normalized by microbial biomass C content. Activity of seven enzymes were measured: α-glucosidase (AG), β-glucosidase (BG), cellobiohydrolase (CBH), xylosidase (XYL), phenol oxidase (POX), peroxidase (PER) and N-acetyl glucosamide (NAG). Data are averaged over soil horizons.

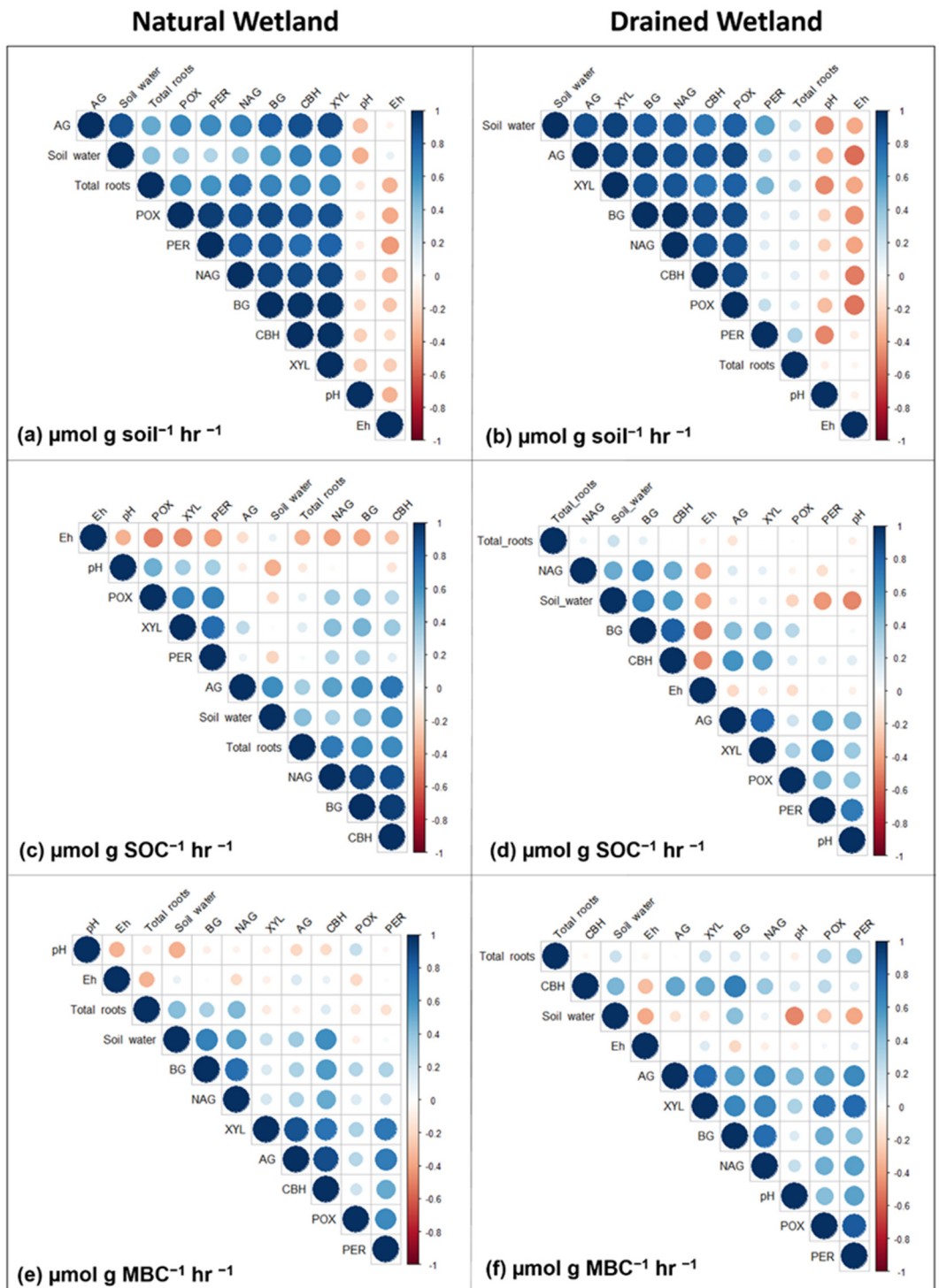

**Figure 5.** Plot showing correlation coefficients (r) between environmental variables and enzyme activity calculated three ways for natural (**a**,**c**,**e**) and drained (**b**,**d**,**f**) forested wetland sites in eastern North Carolina. Activity of seven enzymes were measured: α-glucosidase (AG), β-glucosidase (BG), cellobiohydrolase (CBH), xylosidase (XYL), phenol oxidase (POX), peroxidase (PER) and N-acetyl glucosamide (NAG). The colored gradients at the side of the plot indicate the level of correlation and whether it is positive (blue) or negative (red). The size of the circles also indicates the degree of correlation. Larger/brighter color circles have higher correlation and smaller/faded color circles have lower correlation. Data area averaged over soil horizons.

## 4. Discussion

### 4.1. The Enzyme Activity as Affected by Soil Horizons and Microsites

Microbial extracellular enzyme activity is a crucial step in heterotrophic decomposition of SOC [1,3]. As such, understanding how land-use affects enzyme activity rates is essential to predicting potential changes in SOC storage and cycling, particularly in deep, organic soils of coastal wetlands which store large amounts of C in saturated soils [48]. In this study, we measured the activity of a suite of C- and N-degrading extracellular enzymes and environmental variables across organic and mineral soil horizons to better understand how wetland drainage and spatial variability (microtopography and soil depth) influence enzyme activity, and which C compounds soil microorganisms will breakdown under the different land-use scenarios. We also leveraged previously collected data from the same soil sample collection (Minick et al., 2019) to normalize enzyme rates by SOC and MBC, along with the more widespread normalization by soil weight, in order to elucidate how enzymes normalized using different parameters influenced trends in enzyme activity with drainage and soil depth.

In the organic and mineral soil horizons sampled in this study, activity of hydrolytic and oxidative enzymes fell within the range of the global average of 40 ecosystems [49]. Generally, enzyme activity rates (except PER) were in line with those measured in other freshwater wetlands [36,38,50,51], excluding the Oi which exhibited an order of magnitude higher enzyme activity rate. For PER, activity normalized by SOC showed the greatest differences compared to previous studies, with rates ranging approximately 1–90 $\mu$mol g$^{-1}$ SOC h$^{-1}$ compared to approximately 4 $\mu$mol g OM$^{-1}$ h$^{-1}$ (which would be even lower if we assume half of organic matter is C and adjust Jackson et al. [36] measurements accordingly). Similarly, rates of BG (0.1–8 $\mu$mol g$^{-1}$ SOC h$^{-1}$) and CBH (0.1–2 $\mu$mol g$^{-1}$ SOC h$^{-1}$) normalized by SOC from organic and mineral horizons (excluding the Oi) were similar to those found in a tidal freshwater wetland (BG: 3–13 $\mu$mol g OM$^{-1}$ h$^{-1}$; CBH: < 1 $\mu$mol g OM$^{-1}$ h$^{-1}$; [50]). Rates of BG (0–24 $\mu$mol g$^{-1}$ h$^{-1}$) normalized by soil weight in our study were also similar to those measured in Minnesota peatlands (BG: 2–25 $\mu$mol g$^{-1}$ h$^{-1}$; [51,52]), while activity of POX (0–6.5 $\mu$mol g$^{-1}$ h$^{-1}$) was lower (POX: 6.3–12.7 $\mu$mol g$^{-1}$ h$^{-1}$; [51]). In comparison to surface soil horizons of many upland sites [33,51,53], surface horizons of wetland soils, including those of our sites, have much higher enzyme activity.

Soil depth has been demonstrated to influence enzyme activity in a wide range of upland soils [33,53–55], and to a lesser extent in wetland soils [34,36,51] as conditions affecting enzyme activity (e.g., water table depth, SOC quantity and quality, oxygen, pH, temperature, microbial community structure) shift with soil depth. In our study, soil depth effects on enzyme activity was dependent on how the data were normalized, with exponential decreases in enzyme activity measured with soil depth when normalized per unit soil weight [52,54,55], as compared to either no change or increases with soil depth when normalized per unit SOC or MBC (particularly in the AB and B mineral soil horizons) [33,53]. Higher enzyme activity per unit SOC or MBC at depth may result from proportionally greater enzyme production by soil microorganisms, as well as the presence of relic extracellular enzymes sorbed to soil mineral particles at depth which may desorb from mineral components and interact to decompose soil organic matter [53]. The latter is important in the context of our study, given that specific enzyme activity increased primarily in mineral soils where enzyme interactions with mineral particles would be most likely to occur [56]. Higher pH has also been shown to enhance enzyme activity [57] and we found a positive correlation between pH and enzyme activity rates expressed per unit mass of SOC or MBC but not per unit mass of soil weight (which exhibited a negative correlation with some enzymes). Shifts in microbial community structure with soil depth, and between natural and drained wetlands, may have played a role in determining enzyme activity, but detailed soil microbial community analysis was not an objective of this study.

Soil depth effects on enzyme activity also appear dependent on the type of enzyme, with some studies showing decreases in hydrolytic enzyme activity and no change or increases in oxidative enzymes with depth in northern wetlands, tropical swamp peat soils [34,36] and

arable soils [53]. In our study, hydrolytic enzyme activity in the natural wetland tended to be higher compared to the drained wetland in the organic soil horizons. This may result from the higher SOC content of organic soils of these freshwater forested wetlands (Minick et al., 2019) and high levels of plant and microbial belowground activity [6,24]. We also found an increase in the activity with soil depth of the oxidative enzyme PER when normalized by SOC and MBC. The presence of higher concentrations of lignin at depth in wetland soils [39] would also elicit increased activity of lignin-degrading enzymes, reflecting the different availability of C substrates in subsurface compared to surface soil horizons [53], and PER activity was previously shown to be of importance to wetland C cycling [9].

In regard to oxidative enzymes, previous studies have reported greater activity of enzymes utilizing oxygen as the electron acceptor in drained wetland soils compared to intact, frequently submerged wetland soils [19]. In addition to oxygen concentration, the concentration of phenolic materials (as suggested by 'enzyme latch hypothesis') or iron content (iron can increase the stability of soil organic carbon) can modify enzyme activities. On the contrary, we found that enzyme activity at the natural wetlands tended to be higher than the drained wetland. This could be attributed to the dynamic hydroperiod at our site supplying ample $O_2$ to support the functioning of oxidative enzymes, even at depth within the soil profile where anaerobic conditions have traditionally been presumed to dominate [36]. Peroxidase appears to be an especially important oxidative enzyme in these freshwater wetland soils given its high rate of activity, no change or increases with soil depth [36], enhanced activity under freshwater compared to saltwater conditions [9] and exhibiting the highest activity rates of all enzymes measured in this study. Previous work at our site has suggested that PER activity may also be of importance to methane production as higher rates of PER activity correlated with greater methane production from incubated soils [9,58].

Enzyme activity normalized per unit mass of SOC and MBC was lower in the organic soil horizons of the drained wetland compared to the natural wetland, which was partially contradictory to our hypothesis that drainage would increase microbial activity given previous findings of enhanced organic matter breakdown and microbial activity in drained wetland soils [12,59]. This suggests that microbes are investing more energy into production of extracellular enzymes responsible for degradation of C compounds with fast to slow turnover times in the Oi and Oa soil horizons of the natural wetland, potentially as a mechanism to maximize breakdown of SOC or to increase the likelihood that under stressful environmental conditions at least some minimal amount of C will become available for microbial assimilation. In mineral soils though (AB or B horizon), numerous enzymes (AG, CBH, XYL, POX, PER, and NAG) exhibited higher activity when normalized by unit mass of SOC or MBC in the drained wetland compared to the natural wetland, which is more consistent with previous findings of the positive effects of drainage on decomposition. This may be driven by several factors: (1) the positive effects of a lower water table and increased soil aeration on enzyme activity at depth; (2) greater supply of root-derived organic C from deep rooted pines to microbes in the subsurface soil horizons thereby enhancing enzyme activities targeting organic C and N compounds with faster turnover time (AG, CBH, and XYL) [53,60], and slower turnover time (POX and PER) [61]; and (3) higher pH in the AB and B mineral soil horizons (pH 5.4) of the drained wetland compared to that of the natural wetland (pH 4.8) [16] enhancing microbial enzyme activity [49]. Indeed, we found that enzyme activity tended to be positively correlated with pH when normalized per unit mass of SOC or MBC, particularly for the drained wetland. In support of higher microbial activity in mineral soils of the drained wetland, previous work at our sites showed that SOC in the mineral soil horizons of the drained forested wetland was enriched in $^{13}$C compared to the natural wetland [16], suggesting that this C has been subjected to a greater degree of microbial degradation.

While enzyme rates normalized per unit mass of SOC and/or MBC showed significant differences in organic and mineral soil horizons between the natural wetland and the drained wetland, enzyme activity rates normalized by soil weight showed few differences between sites. Estimating specific enzyme activity rates (either per unit mass of SOC or

MBC) has been shown to be more sensitive to management than those expressed per dry soil weight [33,35]. Our study also found that C-degrading enzyme activities normalized per unit mass of SOC and MBC were more sensitive to drainage and soil depth compared to when normalized by soil weight. For enzyme activity normalized by SOC or MBC, enzymes targeting C compounds with fast (AG and BG) and slow (PER) turnover rates tended to be higher in the Oi and Oa organic soil horizons in the natural wetland compared to the drained wetland. This pattern likely relates to the presence of plant litter in a less decomposed state [16], and therefore containing greater amounts of glucosic and cellulosic substrates compared to SOC in a more decomposed state [62]. Alternatively, the N degrading enzyme NAG was higher in the Oi horizons of the drained wetland compared to the natural wetland, which may reflect the need of microbes to acquire more N from N-poor pine litter compared to more N-rich hardwood litter [63–65].

*4.2. Environmental Effects on Enzyme Activity*

Interestingly, environmental variables showed the most frequent and strongest correlations with enzyme activities normalized by soil weight. One possible explanation is that SOC and MBC inherently reflect soil characteristics such as water holding capacity, C and pH, as they both contribute to and are influenced by these environmental factors. Furthermore, correlations between enzyme activity and total root biomass were almost exclusive to the natural wetland, suggesting the importance of plant-derived C compounds to microbial activity in natural freshwater forested wetlands. This hypothesis is supported by Mitra et al. [66,67], who recently showed that ecosystem-level $CO_2$ and $CH_4$ fluxes were highly correlated with newly acquired plant photosynthate (e.g., GPP) in these natural wetlands. Alternatively, in the drained wetland, the almost complete lack of correlation between enzyme activity rates and total root biomass (except for PER) suggests that in more aerobic systems (such as a drained wetland) belowground plant activity is less of a direct influence on soil microbial activity as the microbes are able to access a wider range of soil organic matter pools compared to natural wetlands. In natural forested wetlands, anaerobic conditions due to high water tables inhibit microbial access to bulk organic matter pools, thereby making fresh C inputs into the soil via root growth, turnover and exudation [24] an important C source to the microbial community, particularly within the rhizosphere where oxygen levels are higher due to actively growing roots.

**5. Conclusions**

Overall, this study highlights the need for a better of understanding of linkages between soil C cycling processes and microbial enzyme activity in wetlands in order to further our understanding of the mechanisms driving wetland C biogeochemistry and storage. In organic horizons, specific enzyme activity (SOC and MBC) (AG, BG, XYL, POX, PER) was higher in the natural compared to drained wetland, but lower (AG, CBH, XYL, POX, PER, NAG) in the AB or B mineral soil horizons. Microtopographical variation at both the natural (hummock-hollow) and drained (bed-interbed) wetlands had little effect on enzyme activity rates. Furthermore, we highlight the need for standardization of enzyme measurements within the field of microbial ecology. Data normalized by soil weight was the least sensitive to management and soil depth, indicating that enzyme rates calculated using soil weight may miss key differences in treatment effects [35]. Data normalized by SOC is an easier metric to measure compared to MBC, with both appearing to be more sensitive to drainage and soil depth effects than normalization by soil weight, providing a more robust and ecologically relevant measurement of soil microbial activity. Results from this study provide useful insights into the biogeochemical functioning of forested wetland soils, particularly looking at broad scale issues of land-use change (managed vs. unmanaged), variation in micro-topography (natural or manipulated), and soil profile characteristics associated with surface and subsurface organic and mineral soil horizons. As coastal wetlands worldwide are under threat from land-use change associated with drainage of wetlands for urban, agricultural, or silvicultural purposes, as well as sea level

rise [10,68,69], it is essential to understand the long-term effects of these disturbances on soil C storage or loss and soil microbial processes which drive soil C cycling within the soil profile. To address these challenges, expansion of such studies to similar sites across the region will increase confidence in the findings and practical application of this study.

**Supplementary Materials:** The following supporting information can be downloaded at: https://www.mdpi.com/article/10.3390/f13060861/s1, Figure S1: Location of field sites in eastern North Carolina, USA. The natural wetland is located in the Alligator River National Wildlife Refuge, Dare County, NC, while the drained wetland (drained for loblolly pine production silviculture) is located in Washington County, NC. Both sites are equipped with an eddy flux tower, with 13 vegetation plots surrounding the tower. Of those thirteen plots, five were utilized for this study (gray shaded circles) (adapted from Minick et al., 2019); Figure S2: Images depicting (A) the natural wetland at Alligator River National Wildlife Refuge and hummock-hollow microtopography (adapted from Minick et al., 2019); and (B) mature loblolly pine plantation in the drained wetland where trees are planted on bedded rows (high microtopographical location) with inter-tree row space (low microtopographical location) (photo credit: Kevan Minick); Figure S3: Soil horizon profile for (A) the natural wetland; and (B) the drained wetland. Letters indicate organic and mineral horizons based on the US soil taxonomical classification system and numbers in parenthesis show the average depth in cm of each horizon.

**Author Contributions:** Conceptualization, K.J.M.; methodology, K.J.M.; software, K.J.M. and M.A.; validation, K.J.M.; formal analysis, K.J.M. and M.A.; investigation, K.J.M.; resources, J.S.K.; data curation, K.J.M.; writing—original draft preparation, K.J.M., J.S.K., B.M. and P.P.; writing—review and editing, K.J.M., X.L., M.A., B.M., P.P. and J.S.K.; visualization, K.J.M. and M.A.; supervision, J.S.K.; project administration, J.S.K.; funding acquisition, J.S.K. All authors have read and agreed to the published version of the manuscript.

**Funding:** Primary funding was provided by USDA NIFA (Multi-agency A.5 Carbon Cycle Science Program) award 2014-67003-22068, DOE NICCR award 08-SC-NICCR-1072, the USDA Forest Service Eastern Forest Environmental Threat Assessment Center award 13-JV-11330110-081, and Carolinas Integrated Sciences and Assessments award 2013-0190/13-2322.

**Data Availability Statement:** Not applicable.

**Acknowledgments:** We thank numerous undergraduate researchers for their invaluable help collecting samples from the field and analyzing samples in the laboratory. We also thank the anonymous reviewers for their comments, which significantly improved the manuscript. The USFWS Alligator River National Wildlife Refuge provided helpful scientific discussions, the forested wetland research site, and valuable in-kind support.

**Conflicts of Interest:** The authors declare no conflict of interest.

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
