# Peer review of "Effects of Spatial Variability and Drainage on Extracellular Enzyme Activity in Coastal Freshwater Forested Wetlands of Eastern North Carolina, USA"

_forests, doi:10.3390/f13060861_

Round 1
Reviewer 1 Report
Dear editor and authors,
I have reviewed the paper. This paper tries to clarify the impact of drainage on enzyme activities. The biggest problem is that this research does not have real replications. The replicated plots are too close to each other, and all of them are in the same location. Therefore, we are unable to know if the impact observed in the present study is really obtained due to drainage.
Minor comments are listed below.
H57:
The response of enzyme activity to drainage has been shown to depend on numerous environmental factors such as the concentration of phenolic and humic compounds, soil pH, O2 levels, and the type of enzyme measured [4,5,18,19,20,21].
It would be better to describe more specifically how these environmental factors control enzyme activity.
L91:
Previous studies have indicated that enzyme activity rates calculated per g dry soil may not be comparable to the specific enzyme activity (enzyme activity per g SOC or MBC) and trends in enzyme activity with soil depth will vary between enzyme type and how enzyme activity is normalized [33,35].
Discussion about this topic (how to express the results of enzymatic activity) is not the main topic in this manuscript, so need to be organized better.
L129- (The natural forested wetland was located at Alligator River National Wildlife Refug…) and L153- (The drained wetland was an intensively managed loblolly pine plantation located in…)
This part is too redundant. I guess it is possible to combine these two paragraphs into one.
L400:
Finally, activity of the N cycling enzyme, NAG, tended to decrease with soil depth regardless of how enzyme rates were normalized (Figure 1G; Figure 2G; Figure 3G), except for the AB mineral soil horizon for data normalized by SOC and MBC in the drained wetland.
The authors regarded NAG as “N cycling enzyme,” but the reaction products also contain C. How can we know if NAG is for N-cycling?
Author Response
Response to Reviewers
We thank both reviewers and the editors for their time and speed in reviewing this manuscript, please see responses below to reviewer 1 comments
Reviewer 1 Comments:
I have reviewed the paper. This paper tries to clarify the impact of drainage on enzyme activities. The biggest problem is that this research does not have real replications. The replicated plots are too close to each other, and all of them are in the same location. Therefore, we are unable to know if the impact observed in the present study is really obtained due to drainage.
Although we do not have multiple sites (spatial replicates) of the forest types being studied, we did choose sites that are very representative of the most common natural and managed forested wetlands within the region based on species composition, biomass/NPP, hydrology, management intensity, etc. Though we did not collect regional data to demonstrate this, the differences between intensively managed pine and unmanaged natural hardwood forests are so drastic (and easily observed) that no one familiar with the region would argue that some (much/most) of the difference is not due to the difference in management intensity. Further, we monitor ecosystem function (GEP, NPP, ERs, SRs, NEP,ET, micro-met, etc.) at both sites continuously over long time intervals (since 2004 at managed and 2009 in natural), and therefore have “temporal replication” that clearly indicates the sites are statistically and consistently different from one another in so many important functional ways. Within each site we have a very large spatial distribution of plots (>500 x 500 m2) which captures the full range of spatial variation (in vegetation, soils, hydrology) that exists within these types of ecosystems. This is not uncommon in studies and we point to other studies that have investigated soil C pools between different wetland types (where only one “site” was used to represent the ecosystems) (see references for examples: Hobbie et al. 2017; Kruger et al. 2014; von Fisher et al. 1995; Domec et al. 2015; Minkkinen and Laine 1998). The results of this study are most applicable when comparing these two sites specifically, but also provide insight on potential impacts of wetland drainage to soil C and microbial pools within similar wetland and land-use types. We agree with the reviewer that lack of spatial replication limits the scope of inference in some ways, but we also trust that readers familiar with large-scale ecological field studies understand that there are unavoidable tradeoffs due to limitations of experimental design that must be addressed in other ways (e.g. long time series, modeling, remote sensing, GIS, etc.). That is, we believe most experienced ecologists will understand the context of our study and the implications for drawing inferences.
We have added a sentence in the discussion to highlight the spatial extent between plots. We have added a sentence in the concluding paragraph underscoring this challenge to long-term, large-scale ecological research and an appeal to expand such studies to other similar sites across the region to increase confidence in the conclusions and practical application of the results.
Minor comments are listed below.
H57:
The response of enzyme activity to drainage has been shown to depend on numerous environmental factors such as the concentration of phenolic and humic compounds, soil pH, O2 levels, and the type of enzyme measured [4,5,18,19,20,21].
It would be better to describe more specifically how these environmental factors control enzyme activity.
We have added some more information regarding how these factors actually influence enzyme activity
L91:
Previous studies have indicated that enzyme activity rates calculated per g dry soil may not be comparable to the specific enzyme activity (enzyme activity per g SOC or MBC) and trends in enzyme activity with soil depth will vary between enzyme type and how enzyme activity is normalized [33,35].
Discussion about this topic (how to express the results of enzymatic activity) is not the main topic in this manuscript, so need to be organized better.
We measured enzyme rates across soil depth in organic and mineral soils and also expressed enzyme activity by three different ways (per unit mass soil vs. per unit mass SOC vs. per unit mass MBC). How enzyme activity rates are calculated does effect the enzyme activities over soil depth, as we discussed in this paragraph. Given that we measured enzyme rates that were expressed in the three different ways and over a range of organic and mineral soils (e.g. soil depth), it seems to the authors that it is relevant to discuss this in the introduction.
L129- (The natural forested wetland was located at Alligator River National Wildlife Refug…) and L153- (The drained wetland was an intensively managed loblolly pine plantation located in…)
This part is too redundant. I guess it is possible to combine these two paragraphs into one.
Both sites are managed quite differently, one by the US National Wildlife Refuge (natural wetland) and one by Weyeraheauser Co (drained wetland) which implores intensive silvicultural management practices and particularly site preparation and bedding which creates the micro topographic variation within this site. The sites also have different histories, different overstory and understory, and different hydrology. Therefore, it seems to the authors that this level of description at each site is warranted to give the reader the most informative description and mental image of the two sites.
L400:
Finally, activity of the N cycling enzyme, NAG, tended to decrease with soil depth regardless of how enzyme rates were normalized (Figure 1G; Figure 2G; Figure 3G), except for the AB mineral soil horizon for data normalized by SOC and MBC in the drained wetland.
The authors regarded NAG as “N cycling enzyme,” but the reaction products also contain C. How can we know if NAG is for N-cycling?
We have removed this wording because it is unclear and confusing.
Reviewer 2 Report
The manuscript “Effects of spatial variability and drainage on extracellular enzyme activity in coastal freshwater forested wetlands of eastern North Carolina, USA” explored the role of drainage, microtopography and soil depth in affecting enzyme activity. Generally, the experiment was well designed and presented, and the topic is interesting and suitable for Forests. The logic is flow and easy to follow, with intensive work and excellent presentation of Figures.
Some very minor corrections:
Line 300: There is two dots at the end of the sentence, please correct.
Line 314: Table 2: It is better to give the full name of AG, BG, CBH... at the bottom of the Table.
Line 621: It seems that the conclusion section is missing, please check.
Author Response
Response to Reviewers
We thank both reviewers and the editors for their time and speed in reviewing this manuscript, please see responses below to reviewer 2 comments
Reviewer 2 Comments:
The manuscript “Effects of spatial variability and drainage on extracellular enzyme activity in coastal freshwater forested wetlands of eastern North Carolina, USA” explored the role of drainage, microtopography and soil depth in affecting enzyme activity. Generally, the experiment was well designed and presented, and the topic is interesting and suitable for Forests. The logic is flow and easy to follow, with intensive work and excellent presentation of Figures.
Some very minor corrections:
Line 300: There is two dots at the end of the sentence, please correct.
fixed
Line 314: Table 2: It is better to give the full name of AG, BG, CBH... at the bottom of the Table.
This information has been added at the bottom of the each table and in the caption for each figure.
Line 621: It seems that the conclusion section is missing, please check.
The last paragraph is the conclusion. We have added the section title.
Round 2
Reviewer 1 Report
Dear editor and authors,
Thank you for the revised version of the manuscript.
Although the authors think preparing replications in the same forest can be acceptable, I could not agree with their opinion.
This is a pseudo-replication. A real replication requires replicated different forests. In the present study, any kind of difference could be significant, just because of "the differences in the two forests."
However, if the editor thinks it is OK, I do not oppose the decision.